# The Nerve Growth Factor Receptor (NGFR/p75^NTR^): A Major Player in Alzheimer’s Disease

**DOI:** 10.3390/ijms24043200

**Published:** 2023-02-06

**Authors:** Francesco Bruno, Paolo Abondio, Alberto Montesanto, Donata Luiselli, Amalia C. Bruni, Raffaele Maletta

**Affiliations:** 1Regional Neurogenetic Center (CRN), Department of Primary Care, ASP Catanzaro, 88046 Lamezia Terme, Italy; 2Laboratory of Ancient DNA, Department of Cultural Heritage, University of Bologna, Via degli Ariani 1, 48121 Ravenna, Italy; 3Laboratory of Molecular Anthropology, Center for Genome Biology, Department of Biological, Geological and Environmental Sciences, University of Bologna, 40126 Bologna, Italy; 4Department of Biology, Ecology and Earth Sciences, University of Calabria, 87036 Rende, Italy

**Keywords:** Alzheimer’s disease, nerve growth factor receptor, NGFR, p75^NTR^, amyloid-beta, expression, signaling pathways, neuropathology, diagnosis, treatment

## Abstract

Alzheimer’s disease (AD) represents the most prevalent type of dementia in elderly people, primarily characterized by brain accumulation of beta-amyloid (Aβ) peptides, derived from Amyloid Precursor Protein (APP), in the extracellular space (*amyloid plaques*) and intracellular deposits of the hyperphosphorylated form of the protein tau (p-tau; *tangles or neurofibrillary aggregates*). The Nerve growth factor receptor (NGFR/p75^NTR^) represents a low-affinity receptor for all known mammalians neurotrophins (i.e., proNGF, NGF, BDNF, NT-3 e NT-4/5) and it is involved in pathways that determine both survival and death of neurons. Interestingly, also Aβ peptides can blind to NGFR/p75^NTR^ making it the “ideal” candidate in mediating Aβ-induced neuropathology. In addition to pathogenesis and neuropathology, several data indicated that NGFR/p75^NTR^ could play a key role in AD also from a genetic perspective. Other studies suggested that NGFR/p75^NTR^ could represent a good diagnostic tool, as well as a promising therapeutic target for AD. Here, we comprehensively summarize and review the current experimental evidence on this topic.

## 1. Introduction

Alzheimer’s disease (AD) represents the most prevalent type of dementia in elderly people [1], primarily characterized by brain accumulation of beta-amyloid (Aβ) peptide in the extracellular space (*amyloid plaques*) and intracellular deposits of paired helical filaments (PHF), mainly formed by the hyperphosphorylated form of the protein tau (p-tau) (*tangles or neurofibrillary aggregates*) [2,3]. The main clinical feature of AD consists of increasing cognitive decline over time, often anticipated, or accompanied by a set of behavioral disturbances [4,5,6]. Two major forms of the disease exist: sporadic (sAD), the causes of which are not yet completely understood and can be related to several hypotheses, and familial (fAD), inherited within families from generation to generation even with a clear autosomal dominant transmission of mutations in Presenilin 1 (*PSEN1*), 2 (*PSEN2)* or Amyloid Precursors Protein (*APP)* genes [7,8,9]. In both forms, amyloid plaques and neurofibrillary tangles are accompanied by neuronal and synaptic loss [10], microgliosis [11], astrogliosis [12], and neuroinflammation [13].

Amyloid plaques originate from the APP, which is a membrane glycoprotein with a large N-terminal glycosylated domain on the extracellular side and a C-terminal domain on the intracellular side of the cell membrane [14]. Under physiological conditions, APP is processed by two distinct pathways, one non-amyloidogenic (α-secretases pathway, e.g., ADAM10 and ADAM17) and one amyloidogenic (β-secretase pathway), the latter of which leads to the production of Aβ peptides (e.g., Aβ_1−42_ and Aβ_25−35_) [15]. Specifically, BACE1 is the β-secretase enzyme that cleaves the transmembrane APP and, together with γ-secretase, generates Aβ species that form increasingly large and conformationally complex soluble aggregates [15,16,17,18]. Over the years, several hypotheses have been formulated to explain how the neuropathological features of AD are causally related to each other, underpinning its pathogenesis. Among these, the most debated in scientific communities is the “amyloid cascade hypothesis” which posed that the progressive accumulation of Aβ in the brain triggers a complex cascade of events that result in the tau hyperphosphorylation, loss of synapses, glial activations, progressive deficiency of neurotransmitters, and neural death [19,20]. However, this hypothesis has been subject to several criticisms or alternative interpretation because of lack of a strong link that can explain how Aβ can cause all these neuropathological phenotypes [21]. One of the most fascinating alternative hypotheses is the “deficient neurotrophic hypothesis”, according to which the loss of trophic support to the cholinergic neurons due to an imbalance of neurotrophins and their receptors underlies the pathogenesis of AD [22,23]. Within this hypothesis, an increasing body of evidence indicates that the nerve growth factor receptor (NGFR), also known as p75 neurotrophin receptor (p75^NTR^), could represent the “missing piece” to elucidate the neuropathological mechanism underpinning both fAD and sAD.

The NGFR/p75^NTR^ protein is encoded by the *NGFR/p75^NTR^* gene, located on the long arm of chromosome 17 (17q21.33). At least two isoforms of NGFR/p75^NTR^ exist: a full-length isoform and a shorts isoform (s-NGFR/p75^NTR^) both of which are expressed in nerve cells [24]. The full-length isoform includes six exons and five introns [25]. The resulting protein comprises an extracellular domain that contains four cysteine-rich motifs (CDR1-CRD4), a transmembrane domain, and an intra-cellular domain formed by a juxtamembrane domain (Chopper) and a death domain (Figure 1) [26,27]. The s-NGFR/p75^NTR^ isoform results from alternative splicing of exon III of the *NGFR/p75^NTR^* locus which encodes for the CDR 2–4 [28] and consequently lacks the neurotrophin-binding site. For this reason, while the full-length isoform can bind all known neurotrophins, the s-NGFR/p75^NTR^ isoform does not interact with several molecules including Nogo, sortilin and Trk (Figure 1) [29,30]. Sabry and colleagues also reported a human NGFR/p75^NTR^ 36 kD isoform, not yet fully characterized [30].

NGFR/p75^NTR^ represents a similar-affinity receptor for all known mammalian neurotrophins—i.e., nerve growth factor (NGF), brain-derived neurotrophic factor (BDNF), NT-3, and NT4/5 [31]-and a higher affinity receptor for their immature forms (i.e., proNGF, pro-BDNF) [32,33]. The ligand binding to NGFR/p75^NTR^ mainly causes the activation of apoptosis signaling pathways [9,34]. However, these neurotrophins can also bind other receptors—i.e., NGF to TrkA, BDNF and NT-4/5 to TrkB, whereas NT-3 to TrkC, promoting downstream signaling involved in neuronal survival and differentiation [35]. Other findings also showed that NGFR/p75^NTR^ can regulate axonal growth, cell cycle, and synaptic plasticity [36]. The specificity of the signaling pathways to be activated (i.e., death vs. survivor) is reached through the formation of NGFR/p75^NTR^ molecular complexes with different co-receptors, such as Trk family, sortilin and Nogo receptors [37]. In particular, the NGFR/p75NTR interaction with Trk receptors induces neuronal survival and differentiation [38,39], the cooperation with sortilin causes the trigger of pro-neutrotrophins-induced cell death [39] whereas the binding with Nogo receptors is involved in the suppression of axonal growth (for a review see [40]).

Moreover, preliminary data indicated that NGFR/p75^NTR^ can also undergo epigenetic modifications such as methylation and histone acetylation. For example, it has been shown that ibuprofen can increase the mRNA and protein levels of NGFR/p75^NTR^ by inducing its promoter hypomethylation and by increasing N^6^-methyladenosine (m6A)-NGFR/p75^NTR^ expression in cancer cell lines [41].

Interestingly, NGFR/p75^NTR^ could also undergo to a three-step proteolytic cleavage that could modify its functional properties [42,43]. During the first step, the extracellular domain (ECD) of NGFR/p75^NTR^ is cleaved by ADAM17, generating a membrane-bound C-terminal fragment (CTF). Then, CTF undergoes to a second cleavage by the PSEN-dependent γ-secretase, releasing the intracellular domain (ICD) into the cytoplasm [31].

Given the multiple functions and implications of NGFR/p75^NTR^ in the central nervous system (CNS), this review will explore the genetic and molecular evidence surrounding its relationship with AD, the experimental evidence around NGFR/p75^NTR^-driven AD neuropathology, as well as the relevance of this protein as a prognostic–diagnostic tool and therapeutic target for AD.

## 2. Expression and Signaling Pathways of NGFR/p75^NTR^ in AD

NGFR/p75^NTR^ receptors are highly expressed of both neurons and glial cells during the CNS development [44] and after brain injury [45], whereas in the normal adulthood are present only in specific brain regions such as neurons and astrocytes of hippocampal cornus ammon 1 (CA1) [46] and especially in the cholinergic neurons (CNs) of the basal forebrain, of which antibodies against these receptors represent the most used immunohistochemical marker [47,48,49]. The CNs of the basal forebrain send long projections to several cortical areas, including the entorhinal cortex and visual cortex, as well as the amygdala, resulting in several cholinergic circuits involved in the regulations of working memory, visual discrimination, and attention [50].

It has long been known that the CNs of the basal forebrain play a critical role in AD for different reasons: (i) the AD neurodegeneration could originate in these neurons and only then may affect other brain regions; (ii) CNs represent the type of neurons most affected in AD; (iii) CNs degeneration predicts the cortical spread of AD neuropathology [51]. Interestingly, TrkA expression is reduced in early and late stages of AD [52,53,54] whereas the NGFR/p75^NTR^ expression is not affected or is increased in AD-damaged CNs of the basal forebrain [54,55,56]. Besides the basal forebrain, AD patients showed higher levels of NGFR/p75^NTR^ receptors also in cortical neurons [57,58], entorhinal cortex [59] and hippocampi [60,61]. Since NGFR/p75^NTR^ mediates apoptotic signaling in the absence of TrkA [45,62] the imbalance ratio of NGFR/p75^NTR^ and TrkA may result in an increased programmed cell death. Sáez and collegues [56] demonstrated that the Aβ_25−35_ exerts an indirect neurotoxic effect by stimulating the astrocytes synthesis of NGF that in turn causes tau hyperphosphorylation and hippocampal neuron death through NGFR/p75^NTR^ [63]. Therefore, NGFR/p75^NTR^ and TrkA receptors differentially regulate APP processing, promoting the non-amyloidogenic and amyloidogenic pathways, respectively. In particular, the binding of NGF to NGFR/p75^NTR^ leads to the activation of the second messenger ceramide and then to the β-secretase cleavage of APP [64].

Interestingly, an increasing amount of evidence indicates the pivotal role of NGFR7p75^NTR^ in amyloidogenesis and in Aβ-induced neurotoxicity due to the binding of Aβ to these receptors. In particular, Devarajan and Sharmila [58] demonstrated that Aβ_1-42_ specifically recognizes CRD1 and CRD2 domains of the NGFR/p75^NTR^ receptor and forms a “cap” like structure at the N-terminal that is stabilized by a network of hydrogen bonds [65].

Moreover, the binding of Aβ_1−42_ to NGFR/p75^NTR^ increases the extracellular levels of Aβ_1−42_ [66], causes neurotoxicity [62] by signaling pathways that involve JNK, NADPH oxidase, and caspases-9/3, as well as by an alternative PLAIDD pathway [67,68,69], mediates Aβ-induced tau hyperphosphorylation and neurodegeneration (i.e., synaptic disorder and neuronal loss) through the calpain/CDK5 and AKT/GSK3β pathways [70] and increases the expression of another pro-apoptotic neurotrophin receptor, sortilin via the RhoA pathway [58] that in turn could interact with NGFR/p75^NTR^ to induce neuronal apoptosis [71]. Chakravarthy and colleagues suggested that Aβ accumulation in turn upregulates the expression of membrane-associated NGFR/p75^NTR^ through insulin-like growth factor 1 receptor (IGF-1R) phosphorylation [61,72]. Moreover, Hu and colleagues reported in 2013 that the death receptor 6 (DR6) forms a receptor complex with NGFR/p75^NTR^ that mediates Aβ-induced neurotoxicity in cortical neurons [73]. Finally, preliminary data also indicated that APP and proNGF could directly interact with NGFR/p75^NTR^ increasing apoptosis in AD [59,74,75] (Figure 2). Moreover, preliminary data indicated that NGFR/p75NTR could facilitate the effects of soluble Aβ oligomers on dendritic spine density and structure in non-apoptotic hippocampal neurons [76].

Interestingly, several data indicated that NGFR/p75^NTR^ ECD, derived from the proteolytic cleavage by ADAM-17 followed by regulated-intramembranous proteolysis by γ-secretase, represents a protective factor against AD-neuropathology [77]. This effect is probable due to the fact that the NGFR/p75^NTR^ ECD contains the binding sites for Aβ—i.e., CRD1 and CRD2 domains—and for other ligands that can induce apoptosis through the activation of these receptors [78]. Thus, the impaired NGFR/p75^NTR^ ECD shedding and the imbalanced levels of NGFR/p75^NTR^ and NGFR/p75^NTR^ ECD could play an important role in regulating the production and clearance of Aβ [79]. Moreover, the *PSEN1* M146V mutation increases γ-secretase cutting of NGFR/p75^NTR^ in vitro and potentially exacerbates the pathogenic outcomes observed in fAD [80]. He and colleagues also reported an overproduction of naturally occurring autoantibodies against NGFR/p75^NTR^ ECD in AD patients, strengthening the role of the immune system and immunosenescence in the AD pathogenesis [79].

Further evidence about the involvement of NGFR/p75^NTR^ in AD derives from inactivation and ablation studies. Indeed, the 5xFAD mice, which expresses a signaling-deficient variant of the NGFR/p75^NTR^, shows a greater neuroprotection from AD neuropathology than animals lacking this receptor [81]. In the same manner, the ablation of NGFR/p75^NTR^ signaling protects the hippocampal network against pathophysiological changes observed in AD (e.g., Aβ-induced degradation of gamma oscillations and gamma–theta interaction) in vitro [82].

Thus, the involvement of NGFR/p75^NTR^ in the AD-associated brain alterations appears to be multifactorial and encompasses several ligands, including Aβ, NGF, proNGF and APP, as well as the activity of ADAM17 and γ-secretases, although the complete set of elements needed by the neurons to enter the apoptotic program in response to these interactions remains to be determined.

## 3. *NGFR/p75^NTR^* Genetic Variants and AD

The first evidence of a genetic involvement of *NGFR/p75^NTR^* in AD emerged from a study of Cozza and colleagues [83]. The authors analyzed the genetic variability of four functional *NGFR/p75^NTR^* polymorphisms (i.e., rs2072445 and rs2072446 located in exon 4, rs741072 located in exon 6, rs734194 located in the 3′-UTR) and the risk of developing AD in an Italian sample consisting of 151 sAD patients, 100 fAD patients, and 97 healthy subjects. fAD was defined as those patients with at least two first-degree relatives in two generations affected by AD in absence of mutations in *PSEN1*, *PSEN2* and *APP* genes. From the results of this study emerged a significant association between the rs2072446 variant and the risk of fAD. Since the polymorphism is located near the genetic region that codes for the cleavage region recognized by ADAM17, its variability may affect the cleavage efficiency of NGFR/p75^NTR^ and, thus, the release of NGFR/p75^NTR^ ECD [79]. However, this association did not remain significant after correction for multiple comparisons. In a second study, Cheng and colleagues [84] examined the genetic variability of the four SNPs previously analyzed by Cozza and collaborators [83] together with the rs741073 variant, located on 3′-UTR, and the risk of developing AD in a Chinese sample consisting of 264 sAD patients and 389 healthy controls the authors found that the rs734194 variant was significantly associated with a decreased risk of AD. However, also in this case, this association did not remain significant after correction for multiple comparisons. Moreover, Matyi and colleagues [85] analyzed the association between the rs2072446 variant and the risk of AD in a population sample of older adults from the Cache County Study on Memory in Aging (CCSMA) of Utah, consisting of 396 AD and 3272 controls. They found a significant association between the genetic variability of the rs2072446 polymorphism and the risk of developing AD only in females. In particular, they found that compared to male non-carriers, female carriers of the minor T allele of rs2072446 showed a 60% higher risk of developing AD in contrast with the finding previously reported by Cozza [83]. Through the analysis of an Australian sample consisting of 258 cases and 247 controls, Vacher and colleagues [86] found a significant association between the variant rs9908234, located in the intron 1 of the *NGFR/p75^NTR^* gene, and the brain accumulation of Aβ. More recently, He and collaborators [87] conducted a case–control association study (366 sAD patients and 390 age-and sex-matched controls) in a Chinese Han population. By analyzing the variability of twelve tag-SNPs within the *NGFR/p75^NTR^* gene located in the promoter region (rs603769 and rs2584665), in intron 1 (rs9908234 and rs3785931), in intron 3 (rs2537706 and rs534561), in exon 4 (rs2072446) and in exon 6 (rs7219709, rs1804011, rs734194, rs741072, and rs741073), they found that the rs2072446-T allele was significantly associated with an increased risk of sAD (OR = 1.79). Interestingly, by analyzing another cohort (279 cognitive normal, 480 MCI, and 47 AD) from the Alzheimer’s Disease Neuroimaging Initiative (ADNI) database it has also been shown that T allele at the rs2072446 variant was also associated with the heavier Aβ burden, which further contributed to an increased risk of AD progression in APOE ε4 non-carrier. Probably the heterogeneity of these results might be due to the presence of sex- and population-specific associations that might be also modified by socio-economic and lifestyle factors and epistatic genetic effects that could be more efficiently captured in large population samples with more accurate and detailed clinical data. Thus, other studies are needed to better understand the role of *NGFR/p75^NTR^* variants in different populations and selecting a set of more informative SNPs to capture most of the genetic variations of the entire gene.

## 4. NGFR/p75^NTR^ as a Biomarker of AD

The evidence that NGFR/p75^NTR^ could represent a specific biomarker for AD emerged in 2015 from a study of Yao and colleagues [77], who reported a significant reduction of NGFR/p75^NTR^ ECD levels in cerebrospinal fluid (CFS) and in the brains of AD patients and of *APP/PSEN1* double-transgenic mice, due to the Aβ-induced reduction in the expression and activity of ADAM17 (Table 1). In the same period, Jiao and collaborators [88] reported a distinct and typical NGFR/p75^NTR^ ECD profile in AD patients, characterized by a decreased CFS and an increased serum levels, compared to other diseases (i.e., Parkinson’s disease, stroke) as well as to the control group (elderly people without neurological disorders). The serum and CFS levels of NGFR/p75^NTR^ ECD also correlated with the Mini-Mental State Examination (MMSE) scores in AD patients and, therefore, its detection could represent a good tool for differential diagnosis of these diseases. In addition, the combination of CSF A_β1-42_, CSF Aβ_42-40_, CSF ptau_181_ or CSF ptau_181_/Aβ_1-42_ with CSF NGFR/p75^NTR^ ECD improves diagnostic accuracy (Table 1). More recently, He and colleagues [79] investigated the presence and alterations of naturally occurring autoantibodies against NGFR/p75^NTR^ ECD in AD patients, as well as their effects on AD pathology. From the results of this study emerged an increased level of these naturally occurring antibodies in the CFS and a negative association between these and the CFS levels of NGFR/p75^NTR^ ECD in AD patients (Table 1). Interestingly, transgenic AD mice actively immunized with NGFR/p75^NTR^ ECD showed a lower level of NGFR/p75^NTR^ ECD and a more severe AD pathology in the brain and worse cognitive functions compared to two different control groups immunized with the reverse sequence of NGFR/p75^NTR^ ECD and phosphate-buffered saline, respectively. Moreover, Crispoltoni and collaborators [89] examined the relationship between the plasma levels of NGF as well as the expression levels of TrkA and NGFR/p75^NTR^ in monocytes of patients with mild cognitive impairment (MCI), mild and severe AD. The authors reported an increased concentration of plasma NGF accompanied by a higher expression of TrkA, but not of NGFR/p75^NTR^, in monocytes from patients with MCI and mild AD. Conversely, in patients with severe AD it has been reported a decreased NGF plasma concentration whereas in the monocytes, the expression of TrkA and NGFR/p75^NTR^ were decreased and increased, respectively, and were associated with caspase 3-mediated apoptosis (Table 1). Therefore, this study reported a plasmatic-NGF and monocytic TrkA and NGFR/p75^NTR^ variation during the progression from MCI to severe AD. However, as far as we know, these studies did not carry out a genetic screening on AD patients to verify the presence of mutations in the *PS1*, *PS2* and *APP* genes or have included patients with no family history of dementia. Further studies are needed to understand whether these findings can be applied to both sporadic and genetic AD patients, taking into account gender differences as well.

## 5. NGFR/p75^NTR^ as a Therapeutic Target for AD

Given the results discussed so far regarding the involvement of NGFR/p75^NTR^ in the neuropathology of AD, since the mid-2000s, several research groups have tested in vivo and in vitro the effect of several molecules capable of blocking or altering the expression of these receptor for the treatment of this currently incurable disease [90]. These include LM11A-31 [91], CATDIKGAEC [92,93] and sulforaphane [94] (Table 2). Furthermore, other researchers have experimented the efficacy of the administration of NGFR/p75^NTR^ ECD in counteracting AD phenotype [70,95].

LM11A-31 is a small non-peptide NGFR/p75^NTR^ ligand able to selectively activate survival signaling and simultaneously inhibit pro-apoptosis pathways [96]. Several animal data indicated that this molecule is able to inhibit Aβ-induced neural death and neuritic degeneration, as well as to prevent tau phosphorylation and misfolding, loss of cholinergic neurites, microglia and astrocyte activation, and cognitive decline and to reverse synaptic function [91,97,98,99,100]. These encouraging results have prompted the researchers to test its effects in humans as well. Indeed, a phase II clinical trial is currently underway on mild–moderate AD patients (ClinicalTrials.gov accession number NCT03069014; Available online: https://clinicaltrials.gov/ct2/show/NCT03069014?term=LM11A-31-BHS&draw=2&rank=1 accessed on 22 December 2022).

In 2007, Yaar and colleagues [92,93] tested the efficacy of CATDIKGAEC heterocyclic peptide, homologous to amino acids 28–36 of NGF, in counteracting the AD phenotype. Preliminary data showed that also this molecule is able to interfere with Ab_1-40_ signaling and rescue neurons from Aβ_1–40_-induced toxicity in vitro. In addition, the in vivo injections of Aβ together with CATDIKGAEC into the cerebral cortex of B57BL/6 mice significantly decreased the Aβ-induced brain inflammation compared to the injection of Aβ and a control peptide. However, as far as we know, this line of research was no longer pursued. Moreover, in 2017, Zhang and colleagues [94] tested the efficacy of sulforaphane (SFN), a secondary metabolite found in edible cruciferous vegetables, in the treatment of AD. Interestingly, the administration of this molecule to *APP/PSEN1* double-transgenic mice, ameliorated the cognitive dysfunction and protected against the Increment of amyloid plaques. These effects may be associated with an up-regulation of NGFR/p75^NTR^ expression which is mediated, at least partly, by SFN-induced reduction in the expression of histone deacetylases (i.e., HDAC1 and HDAC3) that regulates the NGFR/p75^NTR^ transcription. Other studies are needed to better characterize the effects of this molecule in the attenuation of the AD-phenotype, as well as the underlying molecular mechanisms in which NGFR/p75^NTR^ could be implicated.

Finally, as mentioned above, several data indicated that NGFR/p75^NTR^ ECD could have a protective effect on Aβ-induced neuropathology and has been proposed as a good candidate AD biomarker [77]. Interestingly, it has also been shown that the restoration of NGFR/p75^NTR^ levels thought the intracerebroventricular injection of a soluble NGFR/p75^NTR^ ECD-fusion construct protected Tau P301L mice against pathological tau modifications [70] whereas the intramuscular delivery of an AAV-NGFR/p75^NTR^ ECD vector in *APP/PSEN1* double-transgenic mice ameliorates spatial learning and memory, a reduced Aβ accumulation in the brain and blood and reduced neurite degeneration, neuronal death, microgliosis, inflammation, and tau phosphorylation [95]. Therefore, also the use of a NGFR/p75^NTR^-based peripheral approach represents a promising direction to develop novel therapies for AD.

## 6. Conclusions

The evidence here reported, underline the pivotal role of NGFR/p75^NTR^ in all neuropathological changes observed in AD. In the adult brain, these receptors are highly expressed by the cholinergic neurons CNs of the basal forebrain which represent the type of neurons firstly and most affected in AD. Indeed, the binding of Aβ to these receptors increases the extracellular levels of Aβ_1–42_, causes neurotoxicity, synaptic disorder, and neuronal loss, mediates Aβ-induced tau hyperphosphorylation and increases the expression of membrane-associated NGFR/p75^NTR^ as well as of its “death” co-receptors sortilin. The presence of reduced NGFR/p75^NTR^ ECD levels in CFS and in the brains of AD patients, which has a protective effect in AD-like pathology since it prevents the binding of Aβ NGFR/p75^NTR^, not only pushes to consider it as an AD powerful biomarker but opens new avenues for understanding why impaired NGFR/p75^NTR^ signaling occurs in this disease. The genome-wide association studies (GWAS) carried out so far indicate that, only in part, it can be explained by the variability of polymorphisms in the *NGFR/p75^NTR^* gene—currently “only” the rs9908234 and rs2072446 were significantly associated with the brain accumulation of Aβ—although more extensive and large-scale studies are needed in this topic to detect rare NGFR/p75^NTR^ variants and possible epistatic effects with other genetic variants able to modulate the disease susceptibility. For instance, since NGFR/p75^NTR^ ECD is derived from the proteolytic cleavage by ADAM17, further studies are also needed to investigate the role of this biological and functional interaction in the pathogenesis and neuropathology of AD. Moreover, since it has been shown that NGFR promoter is subject to epigenetic regulation, it is also important to clarify the role of the gene expression regulation on the AD susceptibility. Nevertheless, the analysis of the epigenetic variability in relation to the AD onset is still poorly investigated. Finally, the use of NGFR/p75^NTR^ antagonist molecules or the administration of NGFR/p75^NTR^ ECD appear to be promising therapeutic approaches for AD; a phase II clinical trial is currently underway on mild–moderate AD patients with the use of the small NGFR/p75^NTR^ ligand LM11A-31.

## Figures and Tables

**Figure 1 ijms-24-03200-f001:**
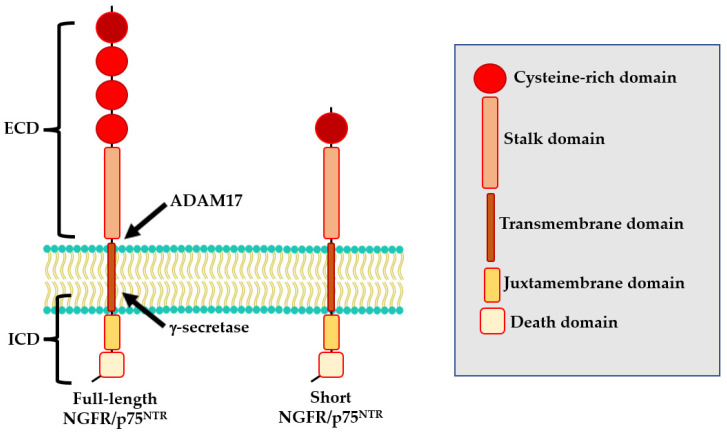
Structure of NGFR/p75^NTR^ isoforms.

**Figure 2 ijms-24-03200-f002:**
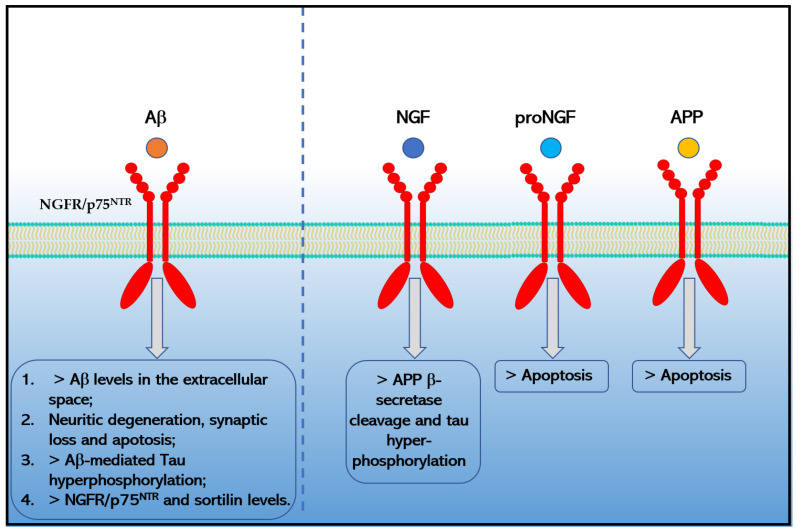
Effects caused by the binding of Aβ_1−42_, NGF, proNGF and APP to NGFR/p75^NTR^ in Alzheimer’s disease.

**Table 1 ijms-24-03200-t001:** Summary of the main studies reporting NGFR/p75^NTR^ as a biomarker for AD.

Authors	Year	Journal	DOI	Comments
Yao et al. [77]	2015	*Molecular Psychiatry*	https://doi.org/10.1038/mp.2015.49	Reduction of NGFR/p75^NTR^ ECD levels in cerebrospinal fluid (CFS) and in the brains of AD patients and of *APP/PSEN1* double-transgenic mice.
Jiao et al. [88]	2015	*Translational Psychiatry*	https://doi.org/10.1038/tp.2015.146	Decreased CFS and an increased serum levels; correlated with the Mini-Mental State Examination (MMSE) scores in AD patients.
Crispoltoni et al. [89]	2017	*Journal of Alzheimer’s Disease*	https://doi.org/10.3233/JAD-160625	Plasmatic-NGF and monocytic TrkA and NGFR/p75^NTR^ variation during the progression from MCI to severe AD.
He et al. [79]	2022	*Neuroscience Bullettin*	https://doi.org/10.1007/s12264-022-00936-4	Negative association between autoantibodies and the CFS levels of NGFR/p75^NTR^ ECD in AD patients.

**Table 2 ijms-24-03200-t002:** Summary of the main studies reporting NGFR/p75^NTR^ as a therapeutic target for AD.

Molecule	Author	Year	Journal	DOI
LM11A-31	Massa et al. [96]	2006	*Journal of Neuroscience*	https://doi.org/10.1523/JNEUROSCI.3547-05.2006
	Yang et al. [91]	2008	*PLoS One*	https://doi.org/10.1371/journal.pone.0003604
	Knowles et al. [97]	2013	*Neurobiology of Aging*	https://doi.org/10.1016/j.neurobiolaging.2013.02.015
	Simmons et al. [98]	2014	*PLoS One*	https://doi.org/10.1371/journal.pone.0102136
	Yang et al. [99]	2020	*Scientific Reports*	https://doi.org/10.1038/s41598-020-77210-y
	Yang et al. [99]	2020	*Acta Neuropathologica Communications*	https://doi.org/10.1186/s40478-020-01034-0
CATDIKGAEC	Yaar et al. [92]	2007	*Neutopathology and Applied Neurobiology*	https://doi.org/10.1111/j.1365-2990.2007.00844.x
	Yaar et al. [93]	2008	*Cellular and Molecular Neurobiology*	https://doi.org/10.1007/s10571-008-9298-6
Sulforaphane	Zhang et al. [94]	2017	*Frontiers in Aging Neuroscience*	https://doi.org/10.3389/fnagi.2017.00121

## Data Availability

Not applicable.

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
