# Peer review of "The Nerve Growth Factor Receptor (NGFR/p75NTR): A Major Player in Alzheimer’s Disease"

_ijms, 2023, doi:10.3390/ijms24043200_

Round 1

Reviewer 1 Report

This is a well written and comprehensive review on the role of NGFR/p75NTR in Alzheimer’s disease.  The authors did an excellent job in reviewing the literature on this topic. There are only few typos that need to be fixed, but other than that it can be accepted without major changes.

Author Response

COMMENT. This is a well written and comprehensive review on the role of NGFR/p75NTR in Alzheimer’s disease.  The authors did an excellent job in reviewing the literature on this topic. There are only few typos that need to be fixed, but other than that it can be accepted without major changes.

RESPONSE. Thank you for this suggestion. We tried to eliminate the typos.

Reviewer 2 Report

The review describes the role of Nerve growth factor receptor (NGFR/p75NTR) in Alzheimer’s disease including also the function of its genetic variations. Moreover, the NGFR probable function as biomarkers and therapeutic target in AD was reported.

The review is quite organized and the covered content is comprehensive. Although, I recommend the publication of the manuscript, I suggest the following minor revisions:

-          The resolution of the figure 2 should be increased. Moreover, the writing should be the same in all the captions present in the figure.

-          The references should be increased with recent publications. Less to 35% of the referees have been published in the last 5 years.

Author Response

COMMENT. The review describes the role of Nerve growth factor receptor (NGFR/p75NTR) in Alzheimer’s disease including also the function of its genetic variations. Moreover, the NGFR probable function as biomarkers and therapeutic target in AD was reported.

The review is quite organized and the covered content is comprehensive. Although, I recommend the publication of the manuscript, I suggest the following minor revisions:

Comment 1. The resolution of the figure 2 should be increased. Moreover, the writing should be the same in all the captions present in the figure.

Response 1. Thank you for this comment. We have increased the size of the writing in the boxes as much as possible and standardized the size.

Comment 2. The references should be increased with recent publications. Less to 35% of the referees have been published in the last 5 years.

Response 2. Thank you for this comment. Since this is a summary review, we have included all the articles we could find on this topic regardless of the year of publication. In any case, we have added those few articles that we had missed including those suggested by the reviewer 3.

Reviewer 3 Report

Bruno et al present a Review on the role of the NGFR/p75NTR as a key player in Alzheimer’s Disease. This is a well-organized and well-presented review, with an updated overview on the topic. In general, the references are complete, with some exception (see comments below).

The English language is in general fine, with some minor typos – I then suggest a careful reading to correct them.

Specific comments:

1.      In general, I suggest substituting the “NGFR/p75NTR” nomenclature with only one of the two names, to increase clarity. Probably the second name is the best, as the NGFR could easily be confused with the NGF molecule in a fast reading.

2.      I would suggest indicating the precursor of NGF as “proNGF” instead of “Pro-NGF”, as the former is more widely used in the neurotrophins’ field.

3.      In section “1. Introduction”, I think that few words would be needed to the other hypothesis for Alzheimer’s Disease, besides the Amyloid hypothesis, as for example the loss of trophic support to the cholinergic neurons due to an imbalance of neurotrophins and their receptors. The involvement of p75NTR as a neurotrophin receptor is strictly linked to this.

4.      The paragraph on p75NTR at the beginning of page 3 shall contain a more precise description of the protein in relation to its neurotrophic factor binders. For example, the view of p75NTR as a low-affinity receptor for neurotrophins is not the current view, given that the overall picture is more complex, with an interplay between p75NTR, Trk receptors and sortilin/SorLA receptors, and the involvement of both mature and pro-neurotrophins. In particular, some key literature references in the field are missing on the early identification of the p75NTR protein, as well as some more updated ones. In this whole last part, many key references on the role of p75NTR are missing, including those of the group of Moses V. Chao and Barbara L. Hempstead, two of the main groups working in the field.

5.      In different parts of the review, the sortilin receptor is not properly described. This is a sorting receptor, involved in many processes. In the last 20 years it was found to be a key player in neurotrophins’ biological function and was found to be involved in the pro-neurotrophins mediated apoptosis, in conjunction with p75NTR. This is not reported correctly in the review. Please address this point.

6.      To increase clarity of reading and understanding, I would recommend adding one table in section “3. NGFR/p75NTR genetic variants and AD”, to summarize the polymorphisms, their implication, and the related references.

7.      For the same reason, I would suggest adding a Table in section “4. NGFR/p75NTR as a biomarker of AD”, to show in a more schematic way the different studies in support of p75NTR as a biomarker.

8.      And in analogy with these suggestions, I strongly suggest adding a table in section “5. NGFR/p75NTR as a therapeutic target for AD”, to better highlight the small molecules and/or therapeutic strategies involving p75NTR.

9.      Figure 2 – The character of the comments in the boxes below the drawings are too small and cannot be read properly.

10.  In paragraph “2. Expression and signaling pathways of NGFR/p75NTR in AD”, the authors should also discuss other evidences in the expression and regulation of p75NTR in the context of Alzheimer’s Disease, as for example those reporting increased protein levels (for example Saadipour et al, J Neurochem 2018, Podlesniy et al, Am J Pathol, 2006, and others).

11.  It might be useful to also discuss the described role of p75NTR in mediating the negative effects of Aβ 1–42 oligomers (see for example Patnaik et al, Scientific Reports, 2020).

12.  The last part of the Conclusions is somehow confusing, please clarify.

Author Response

COMMENT. Bruno et al present a Review on the role of the NGFR/p75NTR as a key player in Alzheimer’s Disease. This is a well-organized and well-presented review, with an updated overview on the topic. In general, the references are complete, with some exception (see comments below).

The English language is in general fine, with some minor typos – I then suggest a careful reading to correct them.

Response. Thank you for this comment. In the revised version of the manuscript minor errors and typos have been corrected.

Specific comments:

Comment 1. In general, I suggest substituting the “NGFR/p75NTR” nomenclature with only one of the two names, to increase clarity. Probably the second name is the best, as the NGFR could easily be confused with the NGF molecule in a fast reading.

Response 1. We thank the reviewer for spotting this point. Given the general confusion that exists in the literature regarding the name of this receptor (NGFR vs p75NTR) we have decided that the optimal way to rename this protein is NGFR/p75NTR, in agreement with other works, also published in prestigious journals such as Nature (e.g., https://www.nature.com/articles/3800542). We hope you agree.

Comment 2. I would suggest indicating the precursor of NGF as “proNGF” instead of “Pro-NGF”, as the former is more widely used in the neurotrophins’ field.

Response 2. Thank you for this comment. We have renamed pro-NGF to proNGF.

Comment 3. In section “1. Introduction”, I think that few words would be needed to the other hypothesis for Alzheimer’s Disease, besides the Amyloid hypothesis, as for example the loss of trophic support to the cholinergic neurons due to an imbalance of neurotrophins and their receptors. The involvement of p75NTR as a neurotrophin receptor is strictly linked to this.

Response 3. We thank the referee for this suggestion. The introduction section now also includes a brief description of this hypothesis (marked in red).

 Comment 4. The paragraph on p75NTR at the beginning of page 3 shall contain a more precise description of the protein in relation to its neurotrophic factor binders. For example, the view of p75NTR as a low-affinity receptor for neurotrophins is not the current view, given that the overall picture is more complex, with an interplay between p75NTR, Trk receptors and sortilin/SorLA receptors, and the involvement of both mature and pro-neurotrophins. In particular, some key literature references in the field are missing on the early identification of the p75NTR protein, as well as some more updated ones. In this whole last part, many key references on the role of p75NTR are missing, including those of the group of Moses V. Chao and Barbara L. Hempstead, two of the main groups working in the field.

Response 4. Thank you for this comment. We have updated this information and mentioned the two research groups suggested by the reviewer (marked in red).

Comment 5. In different parts of the review, the sortilin receptor is not properly described. This is a sorting receptor, involved in many processes. In the last 20 years it was found to be a key player in neurotrophins’ biological function and was found to be involved in the pro-neurotrophins mediated apoptosis, in conjunction with p75NTR. This is not reported correctly in the review. Please address this point.

Response 5. We thank the referee for this comment. The revised version of our manuscript now addresses this important point (marked in red).

Comment 6. To increase clarity of reading and understanding, I would recommend adding one table in section “3. NGFR/p75NTR genetic variants and AD”, to summarize the polymorphisms, their implication, and the related references.

Response 6. Thank you for this comment. Unfortunately, the studies carried out so far have shown that only the variants rs9908234 and rs2072446 were significantly associated with the brain accumulation of Aβ. For this reason, in your opinion it is superfluous to introduce a new table with only two polimorphisms. We specified this result better in the conclusions to increase clarity of reading and understanding.

Comment 7. For the same reason, I would suggest adding a Table in section “4. NGFR/p75NTR as a biomarker of AD”, to show in a more schematic way the different studies in support of p75NTR as a biomarker.

Response 7. Thank you for this suggestion. The revised version of our manuscript now includes this additional table.

 Comment 8. And in analogy with these suggestions, I strongly suggest adding a table in section “5. NGFR/p75NTR as a therapeutic target for AD”, to better highlight the small molecules and/or therapeutic strategies involving p75NTR.

Response 8. Again, we thank the referee for this suggestion. The revised version of our manuscript now includes also this additional table.

Comment 9. Figure 2 – The character of the comments in the boxes below the drawings are too small and cannot be read properly.

Response 9. We thank the reviewer for spotting this error. We have increased the size of the writing in the boxes as much as possible.

Comment 10.  In paragraph “2. Expression and signaling pathways of NGFR/p75NTR in AD”, the authors should also discuss other evidences in the expression and regulation of p75NTR in the context of Alzheimer’s Disease, as for example those reporting increased protein levels (for example Saadipour et al, J Neurochem 2018, Podlesniy et al, Am J Pathol, 2006, and others).

Response 10. We thank the reviwer for this suggestion. We had already mentioned the fact that some authors have found high levels of this receptor in the AD brain:

“Interestingly, TrkA expression is reduced in early and late stages of AD [47–49] whereas the NGFR/p75NTR expression is not affected or is increased in AD-damaged CNs of the basal forebrain [49–51]. Besides basal forebrain, AD patients showed higher levels of NGFR/p75NTR receptors also in cortical neurons [52] and hippocampi [53,54]”.

However, the revised version of our manuscript now includes the two references suggested by the reviewer.

Comment 11. It might be useful to also discuss the described role of p75NTR in mediating the negative effects of Aβ 1–42 oligomers (see for example Patnaik et al, Scientific Reports, 2020).

Response 11. Thank you for this suggestion. We have now added this information (marked in red).

Comment 12. The last part of the Conclusions is somehow confusing, please clarify.

Response 12. We thank the reviewer for this comment. We have added a sentence on the 2 NGFR/p75NTR genetic variants associated with an increased accumulation of beta-amyloid. We honestly do not understand which other parts in the conclusions section are not clear ( No other comments by the other 2 reviewers have been reported on this section).